# Factors predicting the visual outcome of intracorneal ring segment for keratoconus

**Apisit Khanthik**[1], **Ngamjit Kasetsuwan**[1,2]*, **Sasi Yaisawang**[1], **Usanee Reinprayoon**[1,2], **Vilavun Puangsricharern**[1,2], **Vannarut Satitpitakul**[1,2]

**1** Faculty of Medicine, Department of Ophthalmology, Chulalongkorn University, Bangkok, Thailand, **2** King Chulalongkorn Memorial Hospital and Faculty of Medicine, Department of Ophthalmology, Excellence Center of Cornea and Limbal Stem Cell Transplantation, Chulalongkorn University, Bangkok, Thailand

* ngamjitk@gmail.com

## Abstract

### Objectives

To identify predictive factors and to construct predictive models using epidemiological and clinical preoperative factors for the visual acuity change after intracorneal ring segment (ICRS) implantation in patients with keratoconus.

### Methods

The medical records of 287 keratoconic eyes of 230 patients implanted with ICRS at Chula Refractive Surgery Center of a tertiary university hospital (Bangkok, Thailand) between January 2012 and March 2022 were retrospectively reviewed for epidemiological and clinical preoperative variables, including those derived from Scheimpflug tomography. After randomly excluding one eye for each bilateral case, the remaining 230 eyes were randomized into two groups: a training group (184 eyes) and a validation group (46 eyes). In the training group, the correlation between the interesting variables and postoperative uncorrected and corrected distance visual acuity change (ΔUDVA and ΔCDVA; logMAR scale) at 6 months was explored, and then the multiple linear regression analysis was used to develop the predictive models. The obtained models were tested using the validation group.

### Results

There were 5 and 14 preoperative variables that statistically correlated with ΔUDVA and ΔCDVA respectively. Only the preoperative corrected distance visual acuity (CDVAp) strongly correlated with ΔCDVA (Beta = -0.746). Using multiple regression, the preoperative uncorrected distance visual acuity (UDVAp) and front mean keratometry were selected in the proposed model for ΔUDVA (adjusted $R^2$ = 38.8%), while the CDVAp and index of surface variance (ISV) were selected in the model $\Delta \mathrm{CDVA} = -0.043 - 0.613 \times \mathrm{CDVAp} + 0.002 \times \mathrm{ISV}$ (adjusted $R^2$ = 48.9%). The ΔUDVA and ΔCDVA models were correct in 47.83% and 63.4% of the validation group within 0.20 logMAR, respectively.

**Data Availability Statement:** All relevant data are within the manuscript and its Supporting Information files.

**Funding:** The authors received no specific funding for this work.

**Competing interests:** The authors have declared that no competing interests exist.

## Conclusions

Potential predictive factors and models for ICRS-induced changes in visual acuity are proposed as adjunctive tools for clinicians. Such tools could be used for case selection and during counselling before ICRS implantation to maximize surgical outcomes.

## Introduction

Keratoconus is an ectatic corneal disorder characterized by progressive biomechanical corneal instability. This can lead to corneal apex thinning, corneal protrusion, irregular astigmatism, and—sometimes—central corneal scarring [1]. Keratoconus is generally bilateral and asymmetrical. Typical onset is during puberty, with progression for around 40 years. Spectacles and rigid contact lenses were the initial treatments used for keratoconus. With progression, this condition can cause irreversible visual loss, leading to corneal transplantation in 10–20% of cases [2]. A reversible and less-invasive surgical option than corneal transplantation is intrastromal corneal ring segment (ICRS) implantation. The ICRS exerts an arc-shortening effect on the corneal architecture, flattening the cornea while improving astigmatism and contact lens tolerance [3–5]. Several studies have emphasized the advantages of ICRS, including its removability, stability, and security, by eliminating the need for an intraocular procedure [6–9].

The corneal response to ICRS in keratoconus remains unpredictable. ICRS improves both uncorrected (UDVA) and corrected visual acuity (CDVA) [9–12]. However, some patients, particularly those whose preoperative corrected visual acuity (CDVAp) was not compromised, experienced worse vision post-implantation [13]. Furthermore, postoperative changes in vision do not regularly correlate with geometric improvements in the cornea [14]. Numerous studies have attempted to determine the relationship between various factors and surgical outcomes to avoid unsatisfactory results [4,7,11,13,15–26]. However, the impact of these factors on ICRS-induced visual outcomes remains inconclusive, and some factors reportedly increase the risk of post-implantation complications. For example, atopic dermatitis has been associated with increased ICRS extrusion [27].

In this study, we aimed to identify predictors of visual outcomes at 6 months (or within 5–12 months) following ICRS implantation. We also created a mathematical model using the abovementioned factors to predict postoperative visual acuities quantitatively. In the future, this model may facilitate patient selection and aid preoperative patient education efforts.

## Materials and methods

### Study subjects

The study protocol was approved by the Institutional Review Board (IRB) of the Faculty of Medicine, Chulalongkorn University, Thailand (COA no. 1517/2022 and IRB no. 525/63) and adhered to the tenets of the Declaration of Helsinki. The trial was registered in the Thai Clinical Trial Registry (TCTR20200929001).

This retrospective study was conducted during November 2022 by collecting data from the medical records of patients who underwent ICRS implantation at the Chula Refractive Surgery Center, King Chulalongkorn Memorial Hospital, Bangkok, Thailand. Although the authors had access to information that could identify individual patients, the data in the recording forms were anonymized to maintain patients' confidentiality. The primary objective of this study was to identify predictors of postoperative visual acuity change 6 months (or within

5–12 months) after ICRS implantation in patients with keratoconus. The medical records of all eligible patients (351 eyes) were included in this study. The inclusion criteria were as follows: patients with keratoconus implanted with ICRS (Ferrara Ring, AJL, Boecillo, Spain) at the Chula Refractive Surgery Centre between January 2012 and March 2022. The diagnosis of keratoconus was based on characteristic signs noted on slit-lamp examination, corneal topography, and tomography. Patients with contact lens intolerance and without contraindications to ICRS implantation (such as central corneal scarring) became candidates for the surgery. The exclusion criteria were as follows: (i) lack of visual acuity measurement during 5–12 months after ICRS implantation (39 eyes), (ii) absence of preoperative data (14 eyes) except for intraocular pressure (IOP) and autorefraction, which were inevitably unmeasurable in advance keratoconus, and (iii) vision-affecting complication and/or ICRS removal within the follow-up period (11 eyes). Finally, 64 eyes were excluded, resulting in a cohort of 287 eyes of 230 patients. An intraclass Correlation (ICC) assessment was performed to quantify the degree of similarity between the measurements from the two eyes of the same patient. According to the ICC results (S1 Table), both eyes of the same patient could not be included simultaneously in the study to preserve the independence of the observations. After randomly excluding one eye for each bilateral case, the remaining 230 eyes were randomized into two groups: 184 eyes were assigned to a group to develop the predictive equation (training group) and the other 46 eyes to a group to validate the proposed equation (validation group).

## Surgical technique

All included subjects underwent ICRS implantation by one of five corneal specialists using femtosecond laser-assisted tunnel creation under topical anesthesia. The arc length, thickness, and number of ring segments were selected according to a previously described nomogram based on the type of keratoconus, the steepest axis, Q-value, and anterior corneal topographic astigmatism. A disposable suction ring was placed and centered after marking the visual axis. Next, an intrastromal tunnel was created by a laser beam with a spot size of 3 μm while focused on a predetermined depth (75% of corneal thickness) from the anterior corneal surface at a 5.0 optical zone. After creating the main incision, the ICRS was implanted using the complete aseptic technique. Postoperative medications included topical steroids and antibiotics.

## Data collection and outcome measurement

Epidemiological, pre-implantation, and post-implantation data were obtained from patients' medical records. Epidemiological data included each patient's sex, the age at onset, age at ICRS implantation, and history of atopy and eye rubbing. Pre-implantation assessments included uncorrected and corrected distance visual acuities (UDVAp and CDVAp; logMAR), IOP (mmHg) by applanation/noncontact tonometer, spherical power (SPH; D) and cylindrical power (CYL; D) from the autorefractor, and corneal data from Scheimpflug tomography (Pentacam HR, Oculus, Wetzlar, Germany). The Scheimpflug tomography-derived variables included the mean keratometry of the front (Km-front; D) and back surfaces (Km-back; D), maximum keratometry (Kmax; D), anterior corneal astigmatism (CA), Q-value of the front (Q-front) and the back surfaces (Q-back), pachymetry (microns), topographic indices, pachymetric indices, and corneal aberration. Six distances between each pair of the four topographic landmarks (apex, pupil center, thinnest point, and Kmax point) calculated using the Pythagorean Theorem were also analyzed. The aberrometry (from the Zernike analysis) comprised the total root mean square (total rms), root mean square of lower-order aberration (LOA), and higher-order aberration (HOA). Moreover, the K-factor (KF), which was previously reported as a potential predictor of ΔCDVA, was also determined in this study [21]. The KF was

indirectly derived from the Scheimpflug tomography by calculating the multiple products of flattest keratometry and corneal astigmatism of the front corneal surface. The visual outcomes determined by the changes in UDVA and CDVA at six months (or within 5–12 months) after the surgery (ΔUDVA and ΔCDVA) were considered as dependent variables for the models. Holladay's technique was used to transform visual acuity into the logMAR scale for cases in which visual acuity was reported as counting fingers, hand motion, light perception, or no light perception. Complications from ICRS implantation were considered secondary outcomes.

## Statistical analyses

Statistical analyses were performed using SPSS Statistics for Windows, version 21 (IBM Corporation, Armonk, NY, USA). Considering the large sample size, the data distribution was assumed to be normal. For comparisons between the training and validation groups, the chi-square test and unpaired t-tests were used for categorical data and continuous data, respectively. Paired t-tests were used to compare pre- and postoperative visual acuity.

Potential predictors for the model development were chosen based on both clinical and statistical significance. Univariate linear regression analysis was initially performed in the training group to evaluate the correlation between each preoperative variable (independent variable) and the dependent variable by using univariate linear regression. The candidate variables for multivariate regression analysis included (1) the variables with $p < 0.25$ in the univariate analysis and (2) those that were known from the literature review to correlate with post-ICRS visual acuity in keratoconus. Candidate variables that were less significant (lower standardized coefficients) and demonstrated a strong correlation with other candidate variables were excluded until the absence of multicollinearity was confirmed by a variance inflation factor (VIF) $< 5$.

A stepwise method was used to construct the predictive models. Model assumptions were evaluated by analyzing the Durbin-Watson test (to confirm the lack of correlation between errors), mean Cook's distance (to detect influential points or outliers), collinearity tolerance, and VIF. The level of significance was set at $p < 0.05$.

## Results

Among the 230 eyes of 230 patients, there were comprised of 123 right and 107 left eyes. This study included 161 males (70.0%) and 69 females (30.0%). The mean age at onset was 20.29 ± 7.75 years (range: 7–50 years), while the mean age at ring implantation time was 27.04 ± 8.46 years (range: 13–59 years). There were 121 (52.6%) patients with atopy and 140 (60.9%) with frequent eye rubbing. Table 1 summarizes the preoperative numerical data, means, standard deviations, and ranges. The mean duration of postoperative measurement was 7.0 ± 1.4 months (5.0–12.0 months). We observed significant improvement in the UDVA and CDVA from 1.07 ± 0.55 logMAR to 0.73 ± 0.43 logMAR (Snellen equivalent from 20/235 to 20/107, $p < 0.001$) and 0.52 ± 0.42 logMAR to 0.35 ± 0.28 logMAR (Snellen equivalent from 20/66 to 20/45, $p < 0.001$), respectively. The means of ΔUDVA and ΔCDVA were -0.34 ± 0.48 and -0.17 ± 0.35 respectively.

The patient demographic data of the training and validation groups are presented in Table 2, which shows the similarity between these two groups.

## Univariate linear regression analysis

Table 3 summarizes the univariate linear regression analysis between the postoperative outcomes (ΔUDVA and ΔCDVA) and each preoperative variable. Neither ΔUDVA nor ΔCDVA

**Table 1. Mean, standard deviation, minimum, and maximum of the preoperative variables (n = 230 eyes).**

| Variables | Mean | SD | Min | Max | Variables | Mean | SD | Min | Max |
|---|---|---|---|---|---|---|---|---|---|
| UDVAp, logMAR | 1.07 | 0.55 | 0.12 | 2.30 | DAT, mm | 0.70 | 0.39 | 0.13 | 2.94 |
| CDVAp, logMAR | 0.52 | 0.42 | 0.00 | 2.30 | DAP, mm | 0.45 | 0.25 | 0.01 | 1.35 |
| IOP*, mmHg | 10.0 | 2.7 | 3.0 | 19.0 | DAK, mm | 0.88 | 0.81 | 0.06 | 4.27 |
| SPH**, D | -5.97 | 4.95 | -27.00 | 2.37 | DTK, mm | 0.92 | 0.63 | 0.14 | 4.84 |
| CYL**, D | -6.13 | 3.01 | -14.00 | 0.00 | DTP, mm | 1.02 | 0.45 | 0.13 | 2.92 |
| Km-front, D | 51.76 | 5.46 | 40.10 | 68.80 | DPK, mm | 1.25 | 0.93 | 0.09 | 4.59 |
| Km-back, D | -7.84 | 1.06 | -11.35 | -5.35 | ISV | 104.5 | 41.1 | 21.0 | 346.0 |
| K factor, D | 339.97 | 184.76 | 6.10 | 1055.60 | IVA | 0.90 | 0.52 | 0.07 | 4.15 |
| Kmax, D | 62.72 | 8.82 | 47.20 | 88.10 | KI | 1.25 | 0.16 | 0.88 | 2.30 |
| CA, D | 6.05 | 2.97 | 0.10 | 18.20 | CKI | 1.11 | 0.07 | 0.97 | 1.34 |
| Q-front | -1.20 | 0.52 | -2.70 | 0.01 | IHA | 42.0 | 32.7 | 0.1 | 158.3 |
| Q-back | -1.15 | 0.53 | -2.45 | 0.25 | IHD | 0.14 | 0.08 | 0.00 | 0.47 |
| CPT | 470.91 | 40.22 | 329.00 | 595.00 | Rmin | 5.48 | 0.73 | 3.83 | 7.15 |
| CAT | 456.88 | 44.45 | 295.00 | 594.00 | I-S value | 5.48 | 4.92 | -12.98 | 32.33 |
| CTT | 447.03 | 43.61 | 285.00 | 584.00 | RMS total, rms | 13.12 | 6.23 | 2.42 | 48.13 |
| BAD-D | 11.27 | 5.05 | 1.46 | 27.09 | LOA, rms | 12.72 | 6.05 | 2.39 | 46.32 |
| PIavg | 2.49 | 1.05 | 0.62 | 8.08 | HOA, rms | 3.16 | 1.64 | 0.40 | 13.08 |
| ARTmax | 147.9 | 70.9 | 27.0 | 517.0 | | | | | |

* number of subjects = 187 eyes **number of subjects = 203 eyes.

**Abbreviations:** IOP, intraocular pressure; UDVAp, preoperative uncorrected distance visual acuity; CDVAp, preoperative corrected distance visual acuity; SPH, spherical power; CYL, cylindrical power; Km-front, mean keratometry of front surface; Km-back, mean keratometry of back surface; KF, K factor; Kmax, maximum keratometry; CA, anterior corneal astigmatism; Q-front, Q-value of front surface; Q-back, Q-value of back surface; CPT, thickness at the pupil center; CAT, thickness at the apex; CTT, thickness at the thinnest point; BAD D, Belin/Ambrósio enhanced ectasia display; PIavg, average pachymetric progression index; ARTmax, Ambrósio relational thickness maximum; DAT: distance from apex to thinnest point; DAP: distance from apex to pupil center; DAK: distance from apex to maximum keratometry point; DTK: distance from thinnest point to maximum keratometry point; DTP: distance from thinnest point to pupil center; DPK: distance from pupil center to maximum keratometry point; ISV, index of surface variance; IVA, index of vertical asymmetry; KI, keratoconus index; CKI, central keratoconus index; IHA, index of height asymmetry; IHD, index of height decentration; Rmin, anterior minimum sagittal curvature; rms, root mean square; LOA, corneal lower order aberration; HOA, corneal higher order aberration.

significantly correlated with the epidemiological variables. Five variables were significantly correlated with ΔUDVA: UDVAp (Beta = -0.635, $p < 0.001$), CDVAp (Beta = -0.343, $p < 0.001$), SPH (Beta = 0.221, $p = 0.005$), CA (Beta = 0.180, $p = 0.014$) and K-factor (Beta = 0.155, $p = 0.035$).

Fourteen preoperative variables were statistically correlated with ΔCDVA. Only CDVAp demonstrated a strong correlation ($|Beta| > 0.60$) with ΔCDVA. In contrast, the others showed a weak correlation ($|Beta| < 0.40$), including UDVAp, Km-back, Km-front, SPH, average pachymetric progression index (PIavg), thickness at the thinnest point (CTT), Q-back, thickness at the pupil center (CPT), Ambrósio relational thickness maximum (ARTmax), Q-front, thickness at the apex (CAT), Belin/Ambrósio enhanced ectasia display (BAD-D), and Kmax.

## Multivariate linear regression analysis

**Predictive model for ΔUDVA after six months.** The candidate variables included 12 preoperative variables ($p < 0.25$) in univariate linear regression and two variables from literature reviews which were anterior minimum sagittal curvature (Rmin) and index of surface variance (ISV). To avoid multicollinearity, only nine variables were included in the multiple regression analysis. By applying the stepwise method, UDVAp, and Km-front were selected as

**Table 2. Patient demographics and clinical data of training group and validation group.**

| Variables | Training group (n = 184) | Validation group (n = 46) | p |
|---|---|---|---|
| Sex (male, female) | 129, 55 | 32, 14 | 0.943* |
| Atopy (present, absent) | 97, 87 | 24, 22 | 0.947* |
| Eye rubbing (present, absent) | 108, 76 | 32, 14 | 0.177* |
| Age at onset, years | 20.58 ± 7.64 | 19.11 ± 8.13 | 0.250† |
| Age at OR, years | 27.37 ± 8.42 | 25.75 ± 8.58 | 0.247† |
| UDVAp, logMAR | 1.039 ± 0.545 | 1.200 ± 0.576 | 0.078† |
| CDVAp, logMAR | 0.502 ± 0.413 | 0.583 ± 0.470 | 0.247† |
| IOP, mmHg[a] | 10.1 ± 2.8 | 9.8 ± 2.6 | 0.549 |
| Spherical power, D[b] | -5.639 ± 4.927 | -7.258 ± 4.881 | 0.061† |
| Cylindical power, D[b] | -6.159 ± 3.069 | -5.999 ± 2.816 | 0.761† |
| Km-front, D | 51.47 ± 5.36 | 52.92 ± 5.77 | 0.107† |
| Km-back, D | -7.79 ± 1.01 | -8.05 ± 1.21 | 0.138† |
| Kmax. D | 62.22 ± 8.59 | 64.71 ± 9.52 | 0.087† |
| Q-front | -1.177 ± 0.500 | -1.285 ± 0.607 | 0.209† |
| Q-back | -1.121 ± 0.499 | -1.268 ± 0.630 | 0.091† |
| CTT, μm | 449.6 ± 44.0 | 436.6 ± 41.0 | 0.070† |
| Total rms | 12.7201 ± 5.8452 | 14.7210 ± 7.4501 | 0.051† |
| ΔUDVA, logMAR | -0.324 ± 0.292 | -0.395 ± 0.529 | 0.375† |
| ΔCDVA, logMAR | -0.160 ± 0.341 | -0.201 ± 0.407 | 0.488† |
| Follow-up time, months | 6.90 ± 1.36 | 7.32 ± 1.51 | 0.069† |

*Comparisons between two groups were made using Chi-square test.

†Comparisons between the two groups were made using unpaired t-tests.

Values are presented as mean ± SD.

[a]Number of subjects = 148 (training group) and 39 (validation group).

[b]Number of subjects = 162 (training group) and 41 (validation group).

explanatory variables of ΔUDVA. A summary of the result is shown in Table 4. The proposed equation is as follows:

$$\Delta UDVA = -0.413 - 0.556 \times UDVAp + 0.013 \times Kmfront$$

## Predictive model for ΔCDVA after six months

The candidate variables included 19 preoperative variables (p < 0.25) in the univariate linear regression and one variable from the literature reviews (K-factor) [21]. To avoid multicollinearity, only nine variables were included in the multiple regression analysis. Using the stepwise method, CDVAp and ISV were selected as explanatory variables for ΔCDVA. A summary of the result is shown in Table 5. The proposed equation is as follows:

$$\Delta CDVA = -0.043 - 0.613 \times CDVAp + 0.002 \times ISV$$

## Validation of the predictive models

For the predictive model of ΔUDVA, the average absolute residual between predicted UDVA and actual UDVA was 0.271 ± 0.257, and the correct prediction was achieved only in 47.83% of cases within 0.20 logMAR.

**Table 3. Univariate linear regression analysis of the preoperative variables for the postoperative change in uncorrected/corrected distance visual acuities in the training group (n = 184).**

| Variables | ΔUDVA | | ΔCDVA | | Variables | ΔUDVA | | ΔCDVA | |
|---|---|---|---|---|---|---|---|---|---|
| | Beta | p | Beta | p | | Beta | p | Beta | p |
| Sex | 0.133 | 0.073 | 0.081 | 0.274 | BAD-D | -0.069 | 0.354 | -0.196 | 0.008 |
| Atopy | 0.049 | 0.507 | 0.038 | 0.606 | PIavg | -0.088 | 0.235 | -0.232 | 0.002 |
| Eye rubbing | 0.073 | 0.325 | 0.06 | 0.421 | ARTmax | 0.124 | 0.093 | 0.208 | 0.005 |
| Age onset | -0.068 | 0.358 | -0.033 | 0.653 | DAT | -0.039 | 0.603 | 0.059 | 0.424 |
| Age OR | -0.059 | 0.425 | 0.003 | 0.971 | DAP | -0.042 | 0.573 | -0.040 | 0.594 |
| IOP* | 0.045 | 0.590 | -0.039 | 0.638 | DAK | -0.012 | 0.868 | 0.100 | 0.175 |
| UDVAp | -0.635 | < .001 | -0.33 | < .001 | DTK | -0.053 | 0.476 | 0.048 | 0.522 |
| CDVAp | -0.343 | < .001 | -0.715 | < .001 | DTP | -0.062 | 0.400 | 0.056 | 0.448 |
| SPH** | 0.221 | 0.005 | 0.241 | 0.002 | DPK | 0.009 | 0.902 | 0.119 | 0.108 |
| CYL** | 0.034 | 0.668 | -0.083 | 0.296 | ISV | -0.001 | 0.989 | -0.087 | 0.243 |
| Km-front | -0.086 | 0.244 | -0.26 | < .001 | IVA | 0.012 | 0.874 | 0.052 | 0.483 |
| Km-back | 0.060 | 0.415 | 0.27 | < .001 | KI | -0.024 | 0.743 | -0.003 | 0.968 |
| KF | 0.155 | 0.035 | 0.008 | 0.919 | CKI | 0.013 | 0.856 | -0.106 | 0.152 |
| Kmax | 0.023 | 0.759 | -0.146 | 0.048 | IHA | 0.077 | 0.300 | 0.037 | 0.618 |
| CA | 0.180 | 0.014 | 0.06 | 0.42 | IHD | 0.020 | 0.790 | 0.012 | 0.877 |
| Q-front | 0.066 | 0.371 | 0.205 | 0.005 | Rmin | -0.015 | 0.837 | 0.140 | 0.057 |
| Q-back | 0.023 | 0.757 | 0.215 | 0.003 | IS | -0.008 | 0.910 | 0.074 | 0.321 |
| CPT | 0.101 | 0.171 | 0.214 | 0.004 | RMS total | 0.070 | 0.347 | -0.003 | 0.965 |
| CAT | 0.099 | 0.183 | 0.203 | 0.006 | LOA | 0.073 | 0.323 | 0.001 | 0.989 |
| CTT | 0.109 | 0.141 | 0.216 | 0.003 | HOA | 0.010 | 0.897 | -0.054 | 0.469 |

[a] All-dummy variables are coded as 0 for "absent" or 1 for "present" (except for sex: 0 = female, 1 = male).

[b] Correlation is significant at the 0.05 level (2-tailed).

*Number of subjects = 148

**number of subjects = 162.

**Abbreviations:** Beta, standardized coefficients of univariate linear regression; ΔUDVA and ΔCDVA, change in uncorrected/corrected distance visual acuities at 6 months after the surgery; OR, operation date (intracorneal ring insertion); IOP, intraocular pressure; UDVAp, preoperative uncorrected distance visual acuity; CDVAp, preoperative corrected distance visual acuity; SPH, spherical power; CYL, cylindrical power; Km-front, mean keratometry of front surface; Km-back, mean keratometry of back surface; KF, K factor; Kmax, maximum keratometry; CA, anterior corneal astigmatism; Q-front, Q-value of front surface; Q-back, Q-value of back surface; CPT, thickness at the pupil center; CAT, thickness at the apex; CTT, thickness at the thinnest point; BAD D, Belin/Ambrósio enhanced ectasia display; PIavg, average pachymetric progression index; ARTmax, Ambrósio relational thickness maximum; DAT: distance from apex to thinnest point; DAP: distance from apex to pupil center; DAK: distance from apex to maximum keratometry point; DTK: distance from thinnest point to maximum keratometry point; DTP: distance from thinnest point to pupil center; DPK: distance from pupil center to maximum keratometry point; ISV, index of surface variance; IVA, index of vertical asymmetry; KI, keratoconus index; CKI, central keratoconus index; IHA, index of height asymmetry; IHD, index of height decentration; Rmin, anterior minimum sagittal curvature; rms, root mean square; LOA, corneal lower order aberration; HOA, corneal higher order aberration.

For the predictive model of ΔCDVA, the average absolute residual between predicted CDVA and actual CDVA was 0.193 ± 0.150, and the correct prediction was achieved in 63.04% of cases within 0.20 logMAR

## Complications of ICRS implantation

Both intraoperative and postoperative complications in 351 eyes were recorded as secondary outcomes. There was one intraoperative complication in which a corneal perforation at the incision site from a stromal spreader was documented. The eye was successfully re-operated on using a manual technique, approximately seven months after the first surgery. Postoperative complications included segment migration (17 eyes, 4.84%), ICRS-related infection (14

**Table 4. Multivariate regression analysis of selected factors to predict the postoperative change of uncorrected distance visual acuity (ΔUDVA) after ICRS implantation.**

| Variables | Unstandardized Coefficients | | Standardized Coefficients | p-value | Collinearity Statistics | |
|---|---|---|---|---|---|---|
| | B | 95% CI | Beta | | Tolerance | VIF |
| (Constant) | -0.413 | -1.003, 0.177 | | 0.169 | | |
| UDVAp | -0.556 | -0.664, -0.448 | -0.653 | < .001 | 0.925 | 1.081 |
| Km-front | 0.013 | 0.001, 0.025 | 0.136 | 0.035 | 0.925 | 1.081 |

$R^2$ = 39.6%; adjusted $R^2$ = 38.8%; F = 52.137 ($p < 0.001$); mean Cook's distance = 0.007 ± 0.012; Durbin-Watson test = 1.882.

**Abbreviation**: CI, confidence interval; UDVAp, preoperative uncorrected distance visual acuity; Km-front, front mean keratometry; VIF, variance inflation factor. The predictability ($R^2$) was 39.6%, and the adjusted $R^2$ was 38.8%, with F = 52.137 ($p < 0.001$). No influential points or outliers were detected (mean Cook's distance, 0.007 ± 0.012). The independence of the residuals (Durbin-Watson test = 1.882) and the lack of multicollinearity were confirmed.

eyes, 3.99%), corneal melting (3 eyes, 0.85%), peri-annular deposits (2 eyes, 0.56%) and spontaneous fragmentation of the ring segment (1 eye, 0.28%). Twelve eyes underwent ring removal, four of which were due to infection-induced corneal melting. One patient requested ring explantation from his right eye because of concerns about corneal irritation and a mild visual acuity drop postoperatively from CDVA 20/16 to 20/25.

## Discussion

To the best of our knowledge, this is the largest retrospective study of the relationship between ICRS-induced visual acuity changes and various demographic and preoperative factors of patients with keratoconus. The diversity of the studied variable types is another strength of this study. This study is also the first attempt to create an equation to predict ΔUDVA after ICRS implantation. The predictive equation from the multivariate analysis revealed that a poorer preoperative UDVA (larger UDVAp value) and lower value of front mean keratometry value resulted in greater UDVA gain. However, the model had low predictability (adjusted $R^2$ = 38.8%) and low validity (accuracy rate less than 50%).

In our univariate analysis, patients with poorer baseline visual acuity (larger value on the logMAR scale) were expected to demonstrate greater UDVA and CDVA improvement. This finding was consistent with those of previous studies [13,15,18,21]. More myopic spherical power was weakly correlated with greater gain in both UDVA and CDVA (Beta = 0.221 and 0.241, respectively). This was in contrast to the study by Alio et al. [16], who found that patients with better visual acuity gain had less myopia than the other groups. Several

**Table 5. Multivariate analysis of selected preoperative factors to predict the postoperative change of corrected distance visual acuity (ΔCDVA) after ICRS implantation.**

| Variables | Unstandardized Coefficients | | Standardized Coefficients | p-value | Collinearity Statistics | |
|---|---|---|---|---|---|---|
| | B | 95% CI | Beta | | Tolerance | VIF |
| (Constant) | -0.043 | -0.150, 0.064 | | 0.428 | | |
| CDVAp | -0.613 | -0.710, -0.516 | -0.743 | < .001 | 0.896 | 1.116 |
| ISV | 0.002 | 0.001, 0.003 | 0.21 | < .001 | 0.896 | 1.116 |

$R^2$ = 49.5%; adjusted $R^2$ = 48.9%; F = 77.947 ($p < 0.001$); mean Cook's distance = 0.011 ± 0.042.

Durbin-Watson test = 1.815.

**Abbreviation**: CI, confidence interval; CDVAp, preoperative corrected distance visual acuity; ISV, index of surface variance; VIF, variance inflation factor. The predictability ($R^2$) was 49.5%, and the adjusted $R^2$ was 48.9%, with F = 77.947 ($p < 0.001$). No influential points or outliers were detected (mean Cook's distance, 0.011 ± 0.042). The independence of the residuals (Durbin-Watson test = 1.815) and the lack of multicollinearity were confirmed.

parameters derived from Scheimpflug-based tomography demonstrated significant correlations with ΔCDVA; however, only corneal astigmatism and K-factor were significantly correlated with ΔUDVA. Based on the standardized coefficients, patients with a more advanced baseline topography (steeper Km, Kmax; and more prolate Q-value) tended to gain more postoperative CDVA lines. However, our findings differed from those of previous studies, which found that patients with lower mean keratometry values experienced greater visual improvement [7,16,24].

The corneal thickness at the apex, thinnest point, and pupil center were significantly correlated with ΔCDVA in a similar direction, i.e., thinner corneal thickness tended to demonstrate more CDVA improvement. This finding is in contrast to that of Zare et al. [7], who stated that a patient with a thicker cornea (thinnest corneal thickness > 400 μm) would demonstrate greater improvements in UDVA and CDVA six months post-surgery. The relationships between ΔCDVA and all pachymetric indices were also shown in our study. The ICRS-induced visual gain increased in patients with poorer pachymetric indices (higher preoperative BADD, higher PIavg, and lower ARTmax).

In the univariate analysis, no topographic indices were significantly correlated with visual acuity change; however, higher ISV and CKI values and smaller Rmin seemed to correlate with greater improvement in CDVA ($p < 0.25$). The ISV was calculated as the standard deviation of the individual sagittal radii from the mean curvature. A higher ISV is an indicator of increased anterior corneal surface irregularity. CKI is the ratio between the mean radius of curvature in the peripheral zone and the mean radius of curvature in the central zone, whereas Rmin denotes the maximum steepness of the cone [28]. As the CKI increases, the cone of the keratoconic cornea becomes far steeper, and Rmin usually decreased. It can be inferred that the cornea with a more irregular surface and the steeper cone will gain a more favorable CDVA from ICRS implantation. Nonetheless, a previous study had reported in the opposite direction that a lower preoperative ISV and larger Rmin were strongly correlated with a better gain in UDVA and CDVA [24].

None of the aberrometric variables showed a significant correlation with changes in visual acuity in our study. This is in agreement with many previous studies that also failed to demonstrate a significant correlation between aberrometry and visual acuity change after ICRS implantation [15,22,23].

The distances between the corneal apex, thinnest point, pupil center, and Kmax point were also studied to test the hypothesis that longer distances might reflect more cone eccentricity and have a greater effect on visual acuity. However, this study failed to show any significant correlation between these distances and the changes in visual acuity. Gatzioufas et al. also studied the association between visual outcome after ring implantation and the distance from the apex to the thinnest point and from the apex to maximum keratometry. However, they found that there was no statistically significant association [17].

In this study, the age at onset, age at surgery, history of rubbing or atopy, or intraocular pressure did not significantly correlate with ICRS-induced visual acuity changes. Although females seemed to gain better UDVA than males after implantation in the univariate analysis (Beta = 0.133, p = 0.073), this association was disproven in the multivariate analysis, similar to previous results [19].

As noted from the univariate analysis in our study, CDVA improvement was anticipated to be greater in patients with more advanced keratoconus because it is usually associated with poorer baseline visual acuity, higher myopia, steeper keratometry, more prolateness, thinner pachymetry, higher surface irregularity, steeper cone, and higher aberration. However, when we considered the predictive equation for ΔCDVA obtained from the multivariate analysis, it required larger value of CDVAp and smaller value of ISV to get more CDVA improvement.

Therefore, as the keratoconus become more advanced, both CDVAp and ISV tend to be larger and these two variables will compete with the effects in the model on CDVA change opposingly.

Our proposed model for predicting ΔCDVA has both common and different points from the previous studies [21,22]. The variable CDVAp in our model has been reported in the study of Pena-Garcia et al. [21] as the most critical predictor of ΔCDVA at six months; whereas the variable ISV has never been included in any previously reported model. While no keratometric parameters appeared in our model, the previously proposed model included at least one keratometric parameter [21,22]. This may be attributed to the different sets of candidate variables entered into the multiple regression analysis and the differences in the ICRS manufacturers in each study.

This study was limited by its sole focus on visual acuity at approximately six months post-surgery, as this time point appeared to be associated with short-term outcomes. Thus, further research with a longer follow-up period is necessary to identify factors that predict the long-term effects of the ICRS. Another limitation was the high rate of missing data owing to the retrospective nature of this study. We excluded some cases because their medical records lacked pre-defined parameters; therefore, our results may not reflect the real-world outcomes in all patients who underwent ICRS. Despite these limitations, our findings provide an overview of how does each preoperative factor affects the visual outcome of ICRS in keratoconus and guide clinicians in estimating uncorrected and corrected visual acuity after surgery. The proposed model for ΔCDVA estimation can be used to determine the balance between the opposite effects of CDVAp and ISV on ΔCDVA to determine which case will gain CDVA from ICRS. This clinical implication can be applied to patient counselling and the adjustment of the ICRS nomogram. Researchers are also interested in conducting studies on corneal biomechanical properties using a dynamic Scheimpflug analyzer (Corvis ST, Oculus, Wetzlar, Germany). This was based on the hypothesis that biomechanical properties of the cornea can be potential predictors of ΔCDVA following ICRS implantation [22]. Unfortunately, we could not perform the aforementioned study because only a few patients underwent the dynamic Scheimpflug analysis before the ICRS surgery at our center.

## Conclusion

In conclusion, baseline visual acuity, refractive error, and numerous Scheimpflug tomography parameters can predict ICRS-induced changes in visual acuity at six months post-surgery. These predictors, along with the proposed model, could assist with appropriate case selection and the correct timing of ICRS implantation. These parameters may maximize visual gain in patients with keratoconus.

## Supporting information

**S1 Checklist. STROBE statement—Checklist of items that should be included in reports of observational studies.**
(DOCX)

**S1 Table. Intraclass Correlation (ICC) assessment to quantify the degree of similarity between the measurements from the two eyes in the bilateral cases (n = 57).**
(DOCX)

**S1 File. The data spreadsheet.**
(XLSX)

## Acknowledgments

We would like to thank Dollapas Punpanich, M.S., and Yuda Chongpison, Ph.D., M.S., MBA, consultants for statistical analysis, and Editage (www.editage.com) for English language editing.

## Author Contributions

**Conceptualization:** Apisit Khanthik, Ngamjit Kasetsuwan, Sasi Yaisawang.

**Data curation:** Apisit Khanthik, Sasi Yaisawang.

**Formal analysis:** Apisit Khanthik, Ngamjit Kasetsuwan, Sasi Yaisawang.

**Funding acquisition:** Ngamjit Kasetsuwan.

**Investigation:** Apisit Khanthik, Sasi Yaisawang.

**Methodology:** Apisit Khanthik, Ngamjit Kasetsuwan, Sasi Yaisawang.

**Project administration:** Ngamjit Kasetsuwan.

**Resources:** Ngamjit Kasetsuwan, Usanee Reinprayoon.

**Software:** Apisit Khanthik, Sasi Yaisawang.

**Supervision:** Ngamjit Kasetsuwan, Usanee Reinprayoon, Vilavun Puangsricharern, Vannarut Satitpitakul.

**Validation:** Apisit Khanthik, Ngamjit Kasetsuwan, Sasi Yaisawang.

**Visualization:** Ngamjit Kasetsuwan, Sasi Yaisawang.

**Writing – original draft:** Apisit Khanthik.

**Writing – review & editing:** Ngamjit Kasetsuwan, Sasi Yaisawang, Usanee Reinprayoon, Vilavun Puangsricharern, Vannarut Satitpitakul.

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
