## [Decision Letter · Decision Letter 0]

15 Aug 2023

PONE-D-23-16349Factors predicting the visual outcome of intracorneal ring segment for keratoconusPLOS ONE

Dear Dr. Kasetsuwan,

Thank you for submitting your manuscript to PLOS ONE. After careful consideration, we feel that it has merit but does not fully meet PLOS ONE’s publication criteria as it currently stands. Therefore, we invite you to submit a revised version of the manuscript that addresses the points raised during the review process.

We look forward to receiving your revised manuscript.

Kind regards,

Rajiv R. Mohan, Ph.D.

Academic Editor

PLOS ONE

Additional Editor Comments:

Dear authors,

The reviewers and EBM find your MSS appropriate for publication pending adequate addressing of concerns given below. Thank you for choosing PLOSOne journal.

Reviewers' comments:

Reviewer's Responses to Questions

**Comments to the Author**

1. Is the manuscript technically sound, and do the data support the conclusions?

Reviewer #1: Yes

Reviewer #2: Yes

2. Has the statistical analysis been performed appropriately and rigorously? 

Reviewer #1: No

Reviewer #2: I Don't Know

3. Have the authors made all data underlying the findings in their manuscript fully available?

Reviewer #1: Yes

Reviewer #2: Yes

4. Is the manuscript presented in an intelligible fashion and written in standard English?

Reviewer #1: Yes

Reviewer #2: Yes

5. Review Comments to the Author

Reviewer #1: This is an interesting and well - written manuscript.

The primary concern here is the violation of the assumption of independence of observations, a principle that is fundamental to many statistical tests. By including both eyes from the same patient, the independence of data points is jeopardised, potentially leading to an increased risk of Type I errors.

I would like to recommend a few potential solutions to address this issue. You may consider:

1) Including data from only one eye per patient (I recommend this option - 230 eyes from 230 patients).

2) If data from both eyes are essential to your analysis, more complex statistical approaches such as Generalised Estimating Equations (GEE) or Mixed Effects Models can account for the correlation between the two eyes.

3) Performing an Intraclass Correlation (ICC) assessment to quantify the degree of similarity between the measurements from the two eyes.

Given these methodological concerns, I strongly recommend seeking the advice of a professional biostatistician. Their expertise will be crucial in ensuring that your analysis appropriately addresses these issues and strengthens the credibility of your findings.

Reviewer #2: The manuscript by Khanthik et al. performs a retrospective analysis of KC patients that received ICRS implants to determine a predictive model for a patient to receive ICRS implantation. Their analysis included various important components including biological factors and various factors to determine diagnose KC including spherical power, astigmatism, keatometry of front and back surface, and thinnest region of eye. Additionally, their discussion included various studies that both agreed and disagreed with their finds. Finally, they included the limitations of their studies including the short term aspect and limited potential in severe KC cases.

6. PLOS authors have the option to publish the peer review history of their article (what does this mean?). If published, this will include your full peer review and any attached files.

Reviewer #1: No

Reviewer #2: No

---

## [Author Response · Author response to Decision Letter 0]

31 Oct 2023

Review Comments to the Author

Reviewer #1: This is an interesting and well - written manuscript.

The primary concern here is the violation of the assumption of independence of observations, a principle that is fundamental to many statistical tests. By including both eyes from the same patient, the independence of data points is jeopardised, potentially leading to an increased risk of Type I errors.

I would like to recommend a few potential solutions to address this issue. You may consider:

1) Including data from only one eye per patient (I recommend this option - 230 eyes from 230 patients).

2) If data from both eyes are essential to your analysis, more complex statistical approaches such as Generalised Estimating Equations (GEE) or Mixed Effects Models can account for the correlation between the two eyes.

3) Performing an Intraclass Correlation (ICC) assessment to quantify the degree of similarity between the measurements from the two eyes.

Given these methodological concerns, I strongly recommend seeking the advice of a professional biostatistician. Their expertise will be crucial in ensuring that your analysis appropriately addresses these issues and strengthens the credibility of your findings.

Response: Thank you for pointing this out. To enhance the credibility of our findings according to these comments, we have consulted a professional biostatistician following your recommendation. The intraclass correlation (ICC) was explored, and we found that there is high similarity between the measurements from the two eyes. Therefore, we decide to recruit only one eye per patient to eliminate the dependency of observations, so the more complex statistical approaches such as Generalised Estimating Equations (GEE) or Mixed Effects Models are not further mandatory. The results of regression analysis have changed as the newly obtained candidate variables altered. 

Reviewer #2: The manuscript by Khanthik et al. performs a retrospective analysis of KC patients that received ICRS implants to determine a predictive model for a patient to receive ICRS implantation. Their analysis included various important components including biological factors and various factors to determine diagnose KC including spherical power, astigmatism, keatometry of front and back surface, and thinnest region of eye. Additionally, their discussion included various studies that both agreed and disagreed with their finds. Finally, they included the limitations of their studies including the short term aspect and limited potential in severe KC cases.

Response: Thank you

---

## [Decision Letter · Decision Letter 1]

2 Jan 2024

Factors predicting the visual outcome of intracorneal ring segment for keratoconus

PONE-D-23-16349R1

Dear Dr. %Kasetsuwan%,

We’re pleased to inform you that your manuscript has been judged scientifically suitable for publication and will be formally accepted for publication once it meets all outstanding technical requirements.

Kind regards,

Rajiv R. Mohan, Ph.D.

Academic Editor

PLOS ONE

Additional Editor Comments (optional):

Dear authors,

I am happy to inform that your revisions/changes adequately addressed all concerns. The manuscript is accepted for publication. Congratulations!

Reviewers' comments:

Reviewer's Responses to Questions

**Comments to the Author**

1. If the authors have adequately addressed your comments raised in a previous round of review and you feel that this manuscript is now acceptable for publication, you may indicate that here to bypass the “Comments to the Author” section, enter your conflict of interest statement in the “Confidential to Editor” section, and submit your "Accept" recommendation.

Reviewer #3: All comments have been addressed

2. Is the manuscript technically sound, and do the data support the conclusions?

Reviewer #3: Yes

3. Has the statistical analysis been performed appropriately and rigorously? 

Reviewer #3: I Don't Know

4. Have the authors made all data underlying the findings in their manuscript fully available?

Reviewer #3: Yes

5. Is the manuscript presented in an intelligible fashion and written in standard English?

Reviewer #3: Yes

6. Review Comments to the Author

Reviewer #3: WELL WRITTEN AND INNOVATIVE TOPIC. GOOD MODEL FOR THE CASE SELECTION AND TIMING FOR THE ICRS implantation. also verification from a expert biostatistician is required in this case.

7. PLOS authors have the option to publish the peer review history of their article (what does this mean?). If published, this will include your full peer review and any attached files.

Reviewer #3: No

---

## [Editor Report · Acceptance letter]

28 Jan 2024

PONE-D-23-16349R1 

PLOS ONE

Dear Dr. Kasetsuwan, 

I'm pleased to inform you that your manuscript has been deemed suitable for publication in PLOS ONE. Congratulations! Your manuscript is now being handed over to our production team.

Kind regards, 

on behalf of

Dr. Rajiv R. Mohan 

Academic Editor

PLOS ONE